# Measuring Discrimination against Older People Applying the Fraboni Scale of Ageism

**Ágnes Hofmeister-Tóth** [1,*] , **Ágnes Neulinger** [1] and **János Debreceni** [2,*]

1    Marketing Institute, Corvinus University, 1093 Budapest, Hungary; agnes.neulinger@uni-corvinus.hu
2    Hospitality Department, Budapest Business School, 1054 Budapest, Hungary
*    Correspondence: agnes.hofmeister@uni-corvinus.hu (Á.H.-T.); debreceni.janos@uni-bge.hu (J.D.)

**Abstract:** The progressive aging of developed societies, caused by profound demographic changes, brings with it the necessity of confronting the subject of discrimination against older people. In the last 50 years, many scales of ageism have been developed to measure beliefs and attitudes towards older adults. The purpose of our study was to adapt the full Fraboni Scale of Ageism (FSA) to Hungarian language and assess its reliability, validity, and psychometric properties. The sample of the study was representative of the Hungarian population, and the data collection took place online. In our study, we compare the dimensions of the scale with other international studies and present the attitudes and biases of the Hungarian population against the older people. The results of the study indicate that attitudes toward older people are more positive among women, older people, and people living in villages. In this study, we concluded that the Hungarian version of the Fraboni Scale of Ageism is a suitable instrument for both measuring the extent of ageism in the Hungarian population and contributing to further testing the international reliability, validity, and psychometric properties of the Fraboni Scale of Ageism.

**Keywords:** older people; stereotypes; ageism; discrimination; measurements; Fraboni Scale of Ageism; population representative study

## 1. Introduction

The current aging of the world's population is the 21st century's dominant demographic phenomenon, which is unprecedented in human history. Hungary belongs to a group of ageing societies: about one in five people in Hungary are over 65, which is 19% of the population, and the proportion of people over the age of 65 is projected to reach 29% by 2070 [1]. The increasing aging of the population is a new phenomenon that brings new challenges, including the perceptions and expectations of a given society towards older people. This new trend can also trigger negative attitudes among members of society, which can lead to discrimination against older people. According to the research of [2], negative discrimination based on gender and age is the most common phenomenon in the countries of the European Union. According to the Special Eurobarometer 437, in 2015, age discrimination is much more prevalent in the eastern parts of Europe than in Western societies. The exceptions are Poland and France. In Poland, more than two-thirds of respondents stated that discrimination against people aged 55 and over is very rare, rare, or non-existent. In France, on the other hand, more than 50% of those surveyed said that discrimination against people aged 55 and over was very widespread or widespread.

Hungary occupies the 24th place ahead of Bulgaria in the list of 25 European countries. According to 61% of Hungarian respondents, discrimination against the old is very or fairly widespread in the country [3]. Older people representation in the Hungarian media is rather negative, as older age is often associated with illness, decreased mental ability, and low status, which can have serious consequences for older people in various fields of life [4,5]. The paper of [6] highlighted the need to continue developing

ageism measurement tools to estimate ageism or to use other measures, such as census and population-representative polling, to assess the extent and impact of ageism. The purpose of our study was to adapt the full Fraboni Scale of Ageism (FSA) to Hungarian language and assess its reliability, validity, and psychometric properties in a representative sample of the Hungarian population.

## 2. Theoretical Background

### 2.1. The Concept of Ageism

People's attitudes and stereotypes are formed at individual, as well as at societal, levels. Individual characteristics, such as gender, age, socio-economic status, etc., can influence one's way of judging people. Societal context can also shape attitudes towards old age. There are prejudices and stereotypes about old age in some societies, such as that, at old age and at the end of life, people are mostly sick, weak, poor, and lonely. According to Kotter-Grün [7] (p. 170), "negative views of aging in general and negative age stereotypes in particular are omnipresent and have far reaching consequences, such as age discrimination or prejudice towards older adults".

Robert Butler [8] was the first to use the category of 'ageism' (p. 243) to express prejudice and discrimination toward older adults only because they are old [8]. Palmore [9] distinguishes four types of ageism: negative prejudice, negative discrimination, positive prejudice, positive discrimination. Palmore [9] identified nine negative stereotypes that characterize negative prejudices against old people: illness, impotence, ugliness, mental decline, uselessness, isolation, poverty, and depression. Positive ageism is less common— Palmore defined eight positive stereotypes: kindness, wisdom, reliability, wealth, political power, freedom, eternal youth, and happiness [9]. According to Levy and Banaji [10], ageism can be explicit and implicit. Explicit ageism occurs when there is a conscious awareness, intention, or control in the thought, feelings, or actions. Implicit ageism occurs with very little awareness or intention and literally impacts social interactions [10]. Although ageism has a long tradition of research, there has been a strong effort in the international literature to define the concept more precisely. Over the years, researchers proposed a number of different definitions more or less explicitly [11]. Iversen et al. [11] carried out a systematic literature review by analyzing 27 definitions of ageism and proposed a new and more complex alternative definition: "Ageism is defined as negative or positive stereotypes, prejudice and/or discrimination against (or to the advantage of) elderly people on the basis of their chronological age or on the basis of perception of them as being 'old' or 'elderly'" [11] (p. 15). According to Donizetti [12], "this definition is particularly interesting because, beyond emphasizing aspects already well-recognized in the literature, such as the three classic social-psychological components cognitive (stereotypes), affective (prejudice), and behavioral (discrimination) and the conscious (explicit) and unconscious (implicit) dimensions, it underlines the individual (micro-level), social (meso-level), and institutional (macro-level) significance of the phenomenon" (p. 11). Ageism as a concept has gone through various changes, and although it is currently acknowledged that ageism can be directed towards any age group, ageism against older adults has thus far received the most attention [13].

Positive or negative stereotypes perceived by older people can have a beneficial or detrimental effect on people's mental or physical health. According to Levy [14], each of the ageism predictors exert their influence on health through three pathways: psychological, behavioral, and physiological. Several studies have provided evidence for these ageism–health pathways [15–17]. Although ageism may affect other age groups, facts show that the older age group is the group that suffers the most from the harmful consequences of ageism. In recent years, several studies have addressed the detrimental effects of the phenomenon of ageism on the health of older people, e.g., deterioration in the mental health of older people [17], increasing mortality [18], exclusion of older citizens [19], loneliness, and a decline in cognitive abilities [20]. The research of Levy et al. [16] was the first study to identify the economic cost that ageism imposes on the health of older people. The authors

concluded that reducing ageism can not only be financially beneficial to society, but can also have a positive effect on the health of older adults.

*2.2. Measurement of Ageism*

Early measures of opinions and attitudes about aging were one-dimensional constructs related to the research of general opinions about older people. One example is the Old People Questionnaire (OPQ), created by Tuckman and Lorge [21], which measures what misconceptions and stereotypes individuals have about older people. The next scale was developed by Golde and Kogan [22] in 1959 and included 20 qualitative sentence-supplement statements that respondents had to complete [22]. Not long after, in 1961, Kogan created a quantitative version of this scale, the Old People Scale, which was designed to measure general attitudes about older adults, with respondents having to evaluate each statement [23]. Another common measurement tool is the Aging Semantic Differential Scale, developed by Rosenkranz and McNevin [24] in the late 1960s, but this scale has been used primarily in gerontological research. Over the years, academics have developed many different scales to measure various aspects and dimension of ageism.

Recently, two studies have looked at and analyzed the most commonly used scales intended to measure opinions and attitudes towards older people. Ayalon et al. [25] conducted a comprehensive analysis using various databases and identified 106 studies that used one of eleven explicit scales to measure ageism in their research. An explicit study of ageism explores people's thoughts, feelings, and behaviors about older people simply because of their age [26]. Implicit studies, which investigate unconscious hidden behavior, do not show that the focus of the study is on age. Consequently, researchers have no direct control over responses to implicit tests, which are thus devoid of societal expectations [27]. In the domain of ageism, most investigations have used the Implicit Association Tests (IAT) [28]. According to the findings of Ayalon et al. [25], out of the eleven scales, only one met the minimum requirements for psychometric validation of the scales: the Expectation Regarding Aging Scale [29]; however, this scale measures only the stereotype dimension, and, although it is an explicit scale, this fact precludes its use as a comprehensive scale [25]. The authors also suggested that there is a need to further study explicit scales that measure all dimensions of ageism in a more diverse group of countries [25].

The study by Klusmann et al. [30] was another publication that aimed to systematically analyze scales that measure not only ageism but, more broadly, views on aging in general (Voice on Aging, VoA). In studying the literature, the authors selected 89 measures that were grouped along eight dimensions. The eight dimensions are: ecosystem, balance, stability, dynamics, complexity, manifestation, awareness, and time perspective. The cluster analysis of their research data showed that the scales are not homogeneous across dimensions, 90% of the scales examine VoA explicitly at the conscious level and neglect the implicit nature of views on aging, which is central to the birth of stereotypes, and most of the scales do not contain a reference to time. More than two-thirds of the 89 studies analyzed by Klusmann et al. [30] are from the USA and, due to this, most of the scales obviously reflect Western, and primarily North American, views and ideas [31,32]. According to Klusmann et al., two-thirds of the scales were validated with either adults and older adults and, in fact, none of the scales would be suitable for use in other age groups, as age itself and aging mean different things to 70-year-olds and to 30-year-olds or children [30].

Both studies evaluated the Fraboni Scale of Ageism, which is an explicit multidimensional comprehensive measure. According to Klusmann et al. [30] (p. 16), "for the Fraboni Scale of Ageism (1990) the mixture of cognitive but also affective and behavioral manifestation of multidimensional VoA traits is characteristic." For our study, we have chosen the FSA because it measures stereotypes, prejudice, and discrimination, is the only one of the most commonly used scales to include all three dimensions, and has also been used by many international scholars in different cultures.

### 2.3. The Fraboni Scale of Ageism

The Fraboni Scale of Ageism [33] was developed in Canada in the early 1990s and is still one of the most commonly used measurements for ageism research. According to Fraboni et al. [33], previous scales of ageism measured only one component of the ageism dimension: the cognitive component. The Fraboni Scale of Ageism (FSA) is a much more complex tool designed to measure antagonistic and discriminatory attitudes and tendencies to avoid older people. The Fraboni Scale of Ageism (FSA) was developed to measure antagonistic and discriminatory attitudes and the tendency toward avoidance, in order to represent a more complete measure of ageism [34]. The mean score of the FSA allows for a multifaceted measure of attitudes toward older adults, as this scale does not examine attitudes alone, but as a combination of the following elements: *antilocution, avoidance, and discrimination*. The validity of the Fraboni Aging Scale has been examined in several countries: USA [34], Israel [35], Canada [36], Italy [37], Iran [38], and China [39]. In most cases, the researchers used a shortened version of the scale; for example, Fan et al. [39] used only 22 items in a Chinese student sample, Bodner and Lazar [35] used 22 items in an Israeli student sample, and Kutlu et al. [40] used 25 items of the original scale. In most cases, among the reasons for deleting several items from the FSA were a low total correlation of the items or the fact that items were inconsistent with local policies, culture, and family patterns [35,36].

The authors of international studies have concluded that the FSA scale can be used well in countries with different cultures. In Hungary, Kolos et al. [41] used an abbreviated and modified 12-statement version of the Fraboni scale on two convenient samples of students and older people. According to their results, ageism existed in both groups, although its level on the modified and shortened FSA scale was higher in the student sample. The aim of our research was to examine the validity of the full Fraboni scale on a representative Hungarian sample. The purpose of the current study was to adapt the full FSA to Hungarian language, then to assess its reliability and validity in the Hungarian population, and then to determine attitudes toward older people in the Hungarian society.

### 3. Methodology

*Data Collection and Sample*

The data collection took place online in November 2018 and was carried out by a professional market research company. The online sample was representative of gender, age, settlement type, and region for the Hungarian population aged 18–65. The original sample size was 800 people, but, in the case of the Fraboni Ageism Scale, subjects with a high non-response rate were excluded, so that, finally, the answers of 776 people were analyzed. The description of the sample is shown in Table 1.

**Table 1.** Description of the sample.

|  | n | % |
|---|---|---|
| Men | 379 | 48.8 |
| Women | 397 | 51.2 |
| 18–29 years | 173 | 22.3 |
| 30–39 years | 187 | 24.1 |
| 40–49 years | 157 | 20.2 |
| 50–65 years | 259 | 33.3 |
| Primary education | 362 | 46.6 |
| Secondary education | 266 | 34.3 |
| Higher education | 148 | 19.1 |
| Budapest | 139 | 17.9 |
| Town | 403 | 51.9 |
| Village | 234 | 30.2 |
| Middle-Hungarian Region | 235 | 30.3 |
| West-Hungarian Region | 238 | 30.6 |
| East-Hungarian Region | 303 | 39.1 |

This research projects included the evaluation of several scales, but the present analysis only covers the results of the Fraboni Scale of Ageism (Fraboni et al. 1990). The original scale consists of 29 items designed to assess both cognitive and affective components of ageism that have rarely been used in their original form. Our research was conducted using all items of the FSA scale, except for one statement. A translation and back-translation confirmed the appropriateness of the Hungarian translation. The correctness of the translation and linguistic clarity of the statement's phrasing was checked by a pre-survey, which was carried out by the authors to test the overall clarity of the questionnaire using a convenient sample (n = 10). Of the 29 statements, one statement (statement 17 of the FSA scale), which related to the use of a community sports facilities for old people, was omitted because there are no such facilities in Hungary, and the respondents in the pre-survey had difficulty in understanding this statement. No changes were made on the FSA after the pre-survey. Before data collection, we invited six experts to evaluate the content validity of the FSA (Hungarian version), based on relevance, as 1 (highly relevant), 2 (quite relevant), 3 (somewhat relevant), and 4 (not relevant). The content validity (CVI) for the full scale was 0.98, indicating satisfactory agreement among selected experts on the Hungarian version of the FSA. The statements of the Fraboni scale were evaluated on a 5-point Likert scale, which ranged from (1) strongly disagree to (5) strongly agree. Item numbers 8, 12, 14, 21, 22, 23, and 24 are positive statements, and scores are reversed when calculating the total scale score. Total score ranges from 29 to 145. Higher scores mean higher levels of ageism. Furthermore, during the survey, respondents had the opportunity to give an 'I don't know/No answer' response to enable everyone to answer the statements relevant to them. Respondents also answered a battery of sociodemographic questions. Analyzing all of the statements of the Fraboni scale, it can be said that, on average, 94% of the respondents answered each question with a standard deviation of 1.82. Furthermore, Cronbach's alpha for the original FSA scale was 0.86 for the full scale; this value is 0.87 in the current sample.

## 4. Results

### 4.1. FSA Factor Structure and Construct Validity

In order to examine the structure of attitudes toward older people among our representative sample, an exploratory factor analysis with principal component analysis and varimax rotation was carried out after recoding all reverse items (8, 12, 14, 21, 22, 23, 24). The exploratory factor analysis (EFA) was carried out by using SPSS 26.0. software. The Kaiser–Meyer–Olkin (KMO) measure of sampling adequacy (KMO = 0.930) and Bartlett's test of sphericity ($p = 0.001$) reached statistical significance. Unweighted least squared method was applied with Varimax rotation, resulting in a five-factor construct that explained up to 52.515% of the cumulative variance. The scree plot showed a break between the third and fourth factors (Figure 1). Therefore, we chose the three-factor model originally provided in the FSA.

The analysis was repeated, and the number of factors to be extracted was limited to three. Three factors represented 44.49% of the variance, with factors 1, 2, and 3 contributing 29.22%, 9.79%, and 5.47%, respectively. Varimax rotation was used to interpret the factors. The results of the factor analysis are presented in Table 2. Item 2 was removed because the load on each factor was <0.3. Factor 1 was similar to the avoidance factor of the original FSA scale and consisted of 13 items that expressed the intention to avoid older people. This factor was labeled '*Avoidance*' in the present study. Factor 2 was labeled '*Stereotype*' and was somewhat similar to the '*Antilocution*' factor of the original FSA scale. Factor 2 consisted of seven statements describing respondents' beliefs about older people. The third factor consisted of seven items with positive emotions towards older people; therefore, it was labeled as '*Positive emotional attitude*'. This factor, in comparison, was different to the third factor of the original FSA scale, but similar to the third factor found by [34], the '*Affective attitude*' factor (Table 3).

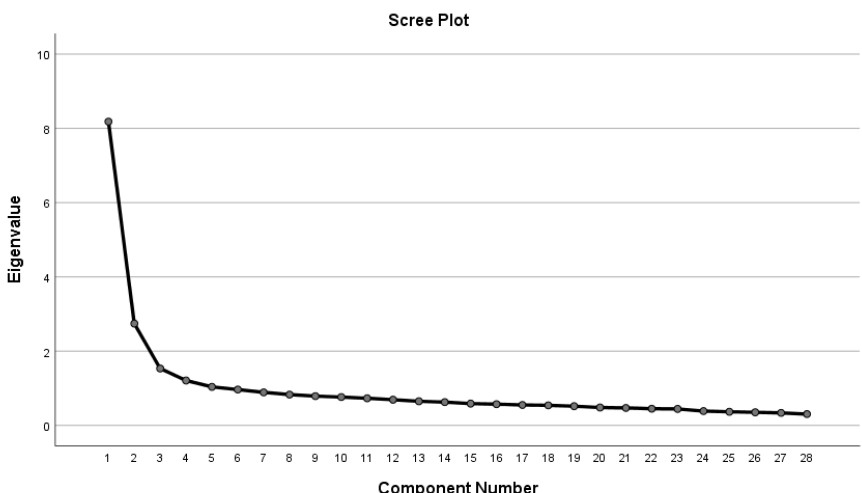

**Figure 1.** Scree plot of exploratory factor analysis.

**Table 2.** Item loadings of the principal component analysis and factor structure.

| Item | Table 1 Item Loading for Principal Component Analysis and Factor Structure | 1 | 2 | 3 | 4 | 5 | 6 | 7 |
|---|---|---|---|---|---|---|---|---|
| 20 | It is best that elderly people live where they won't bother anyone | 0.762 | 0.107 | 0.119 | 1.68 | 1.104 | 1.218 | |
| 6 | I sometimes avoid eye contact with elderly people when I see them | 0.736 | 0.189 | 0.010 | 1.74 | 1.136 | 1.290 | |
| 7 | I don't like it when elderly people try to make conversation with me | 0.669 | 0.242 | 0.117 | 1.93 | 1.187 | 1.409 | |
| 9 | Complex and interesting conversation cannot be expected from most elderly people | 0.605 | 0.324 | 0.220 | 2.04 | 1.217 | 1.481 | |
| 18 | Most elderly people should not be trusted to take care of infants | 0.603 | 0.173 | 0.025 | 2.14 | 1.231 | 1.515 | |
| 15 | I personally would not want to spend much time with an elderly person | 0.602 | 0.326 | 0.182 | 2.23 | 1.244 | 1.548 | 0.840 |
| 25 | Most elderly people would be considered to have poor personal hygiene | 0.598 | 0.291 | 0.020 | 2.00 | 1.177 | 1.386 | |
| 11 | Elderly people should find friends their own age | 0.591 | 0.336 | 0.056 | 2.16 | 1.238 | 1.534 | |
| 26 | I would prefer not to live with an elderly person | 0.510 | 0.380 | 0.193 | 2.52 | 1.329 | 1.766 | |
| 10 | Feeling depressed when around elderly people is probably a common feeling | 0.474 | 0.460 | 0.061 | 2.46 | 1.295 | 1.677 | |
| 13 | I would prefer not to go to an open house at a senior's club if invited | 0.442 | 0.357 | 0.135 | 2.75 | 1.416 | 2.005 | |
| 29 | Elderly people do not need much money to meet their needs | −0.440 | −0.225 | 0.077 | 3.63 | 1.312 | 1.721 | |
| 1 | Teenage suicide is more tragic than suicide among the elderly | 0.302 | 0.298 | 0.053 | 2.70 | 1.548 | 2.396 | |
| 28 | Elderly people complain more than other people | 0.329 | 0.575 | 0.099 | 3.11 | 1.286 | 1.654 | |
| 5 | Many elderly people just live in the past | 0.343 | 0.550 | 0.130 | 3.02 | 1.224 | 1.497 | |
| 4 | Many elderly people are not interested in making new friends, preferring instead the circle of friends they have had for years | 0.112 | 0.525 | 0.028 | 3.33 | 1.197 | 1.433 | |
| 27 | Most elderly people can be irritating because they tell the same stories over and over again | 0.344 | 0.510 | 0.088 | 2.84 | 1.222 | 1.493 | 0.770 |
| 3 | Many elderly people are stingy and hoard their money and possessions | 0.415 | 0.479 | 0.035 | 2.61 | 1.255 | 1.574 | |
| 19 | Many elderly people are happiest when they are with people their own age | 0.336 | 0.412 | −0.086 | 3.12 | 1.216 | 1.478 | |
| 16 | Most elderly people should not be allowed to renew their drivers' licenses | 0.293 | 0.323 | −0.002 | 3.24 | 1.336 | 1.784 | |

**Table 2.** *Cont.*

| Item | Table 1 Item Loading for Principal Component Analysis and Factor Structure | 1 | 2 | 3 | 4 | 5 | 6 | 7 |
|---|---|---|---|---|---|---|---|---|
| 24 | Most elderly people are interesting, individualistic people | 0.032 | 0.107 | 0.738 | 2.24 | 1.075 | 1.155 | |
| 14 | Elderly people can be very creative | 0.029 | 0.188 | 0.710 | 2.37 | 1.115 | 1.243 | |
| 21 | The company of most elderly people is quite enjoyable | −0.054 | 0.151 | 0.605 | 2.51 | 1.075 | 1.156 | |
| 12 | Elderly people should feel welcome at social gatherings of young people | 0.175 | 0.059 | 0.542 | 2.17 | 1.103 | 1.216 | 0.723 |
| 22 | It is sad to hear about the plight of the elderly in our society these days | 0.142 | −0.109 | 0.411 | 1.97 | 1.137 | 1.292 | |
| 8 | Elderly people deserve the same rights and freedoms as other members of our society | 0.349 | −0.188 | 0.386 | 1.41 | 0.869 | 0.756 | |
| 23 | Elderly people should be encouraged to speak out politically | 0.005 | 0.050 | 0.340 | 2.86 | 1.348 | 1.816 | |
| 2 | There should be special clubs set aside within sports facilities so that the elderly can compete at their own level | 0.041 | 0.214 | −0.264 | 3.42 | 1.305 | 1.706 | - |

1 = component: avoidance (loadings); 2 = component: stereotype (loadings); 3 = component: positive emotional attitude factor (loadings); 4 = mean; 5 = SD; 6 = variance; 7 = Cronbach's alpha.

**Table 3.** Comparison of the interpretation of the scale items (item- factor associations) with previous international research.

| Original FSA Statement Number | Present Study Hungary | Fraboni et al. 1990 Canada | Rupp et al. 2005 USA | Bodner-Lazar 2008 Israel | Kutlu et al. 2012 Turkey | Fan et al. 2020 China |
|---|---|---|---|---|---|---|
| 7 | Avoidance | Avoidance | Separation | Stereotype | Discrimination | Avoidance |
| 6 | Avoidance | Avoidance | Separation | Omitted | Avoidance | Avoidance |
| 20 | Avoidance | Discrimination | Separation | Omitted | Dicrimination | Avoidance |
| 15 | Avoidance | Avoidance | Affective attitude | Avoidance | Avoidance | Excluded |
| 9 | Avoidance | Antilocution | Separation | Omitted | Discrimination | Excluded |
| 11 | Avoidance | Avoidance | Separation | Avoidance | Discrimination | Excluded |
| 10 | Avoidance | Avoidance | Separation | Avoidance | Discrimination | Excluded |
| 13 | Avoidance | Avoidance | Stereotypes | Contribution | Avoidance | Excluded |
| 26 | Avoidance | Avoidance | Omitted | Omitted | Avoidance | Avoidance |
| 18 | Avoidance | Discrimination | Stereotype | Avoidance | Stereotypes | Stereotype |
| 25 | Avoidance | Antilocution | Stereotype | Avoidance | Stereotype | Avoidance |
| 1 | Avoidance | Antilocution | Stereotype | Stereotype | Stereotype | Omitted |
| 29 | Avoidance | Antilocution | Omitted | Avoidance | Stereotype | Omitted |
| 27 | Stereotype | Antilocution | Stereotype | Contribution | Stereotype | Stereotypes |
| 4 | Stereotype | Antilocution | Stereotype | Contribution | Stereotype | Stereotype |
| 28 | Stereotype | Antilocution | Stereotype | Omitted | Stereotype | Stereotype |
| 5 | Stereotype | Antilocution | Stereotype | Stereotype | Stereotype | Stereotype |
| 19 | Stereotype | Avoidance | Stereotype | Contribution | Stereotype | Stereotypes |
| 3 | Stereotype | Antilocution | Stereotypes | Stereotype | Stereotype | Stereotype |
| 16 | Stereotype | Antilocution | Omitted | Omitted | Stereotype | Stereotype |
| * 21 | Positive emotional attitude | Discrimination | Affective attitude | Contribution | Avoidance | Excluded |
| * 24 | Positive emotional | Discrimination | Affective attitude | Omitted | Avoidance | Exluded |
| * 14 | Positive emotionattitude | Avoidance | Omitted | Avoidance | Avoidance | Avoidance |
| 12 | Positive emotional attitude | Avoidance | Separation | Contribution | Avoidance | Exluded |
| * 22 | Positive emotional attitude | Discrimination | Affective attitude | Omitted | Omitted | Omitted |

**Table 3.** *Cont.*

| Original FSA Statement Number | Present Study Hungary | Fraboni et al. 1990 Canada | Rupp et al. 2005 USA | Bodner-Lazar 2008 Israel | Kutlu et al. 2012 Turkey | Fan et al. 2020 China |
|---|---|---|---|---|---|---|
| * 8 | Positive emotional attitude | Discrimination | Omitted | Avoidance | Omitted | Omitted |
| 2 | Omitted | Antilocution | Omitted | Omitted | Omitted | Omitted |
| * 23 | Positive emotional attitude | Dicrimination | Affecrive attitude | Avoidance | Discrimination | Avoidance |
| 17 | Omitted | Discrimination | Omitted | Omitted | Omitted | Avoidance |

The three factors of the present study are partially similar to the results of previous international research (see Table 3). * Items 21, 24, 14, 22, 8, 23 were reverse scored.

According to our findings, the average negative perception (the average mean score of the total FSA) of older people among the Hungarian population is moderate (Table 4). Using other Likert scales, but in terms of the average perception of the older generation, Kutlu et al. [40] in Turkey and Donizetti [37] in Italy obtained similar results. Bodner and Lazar [35] in Israel, Sum et al. [38] in Iran, Rupp [34] in the USA, and Allan and Johnson [42] in Canada found above average FSA scores among university students

**Table 4.** Fraboni Scale of Ageism (FSA) means (n = 776).

| FSA Dimensions | Mean | SD | Range | Average Sum of Scores | SD |
|---|---|---|---|---|---|
| Avoidance | 2.309 | 0.750 | 13–61 | 30.01 | 9.736 |
| Stereotype | 3.017 | 0.801 | 7–35 | 21.12 | 5.601 |
| Positive emotional attitude | 2.215 | 0.671 | 7–35 | 15.51 | 4.694 |
| FSA total | 2.502 | 0.578 | 32–128 | 70.06 | 16.195 |

The value of the total FSA score can also be assessed by demographic groups. Based on the ANOVA analysis performed, it can be said that the perception of older adults is statistically significantly more negative among men (x = 2.55, δ = 0.65) than among women (x = 2.41, δ = 0.59). Similarly, there is a difference according to age groups, where the perception of the old generation is the most negative among young people (18–29 years old) (x = 2.72, δ = 0.65), whereas, in the case of the oldest group (50–65 years old), it is the least negative (x = 2.28, δ = 0.52). There is also a statistically significant difference according to the type of settlement, according to which, the perception is more negative in Budapest and in cities (x = 2.52/2.52 and δ = 0.57/0.31, respectively) compared to the opinion of those living in villages (x = 2, 37, δ = 0.63). There was no statistically significant difference by education and region (confidence level for the ANOVA analysis was 95%).

*4.2. Construct Validity*

The three-factor model was tested using ADANCO 2.2.1. software. A partial least square (PLS) technique was performed to examine inter-construct relationships and model characteristics. ADANCO provides several indices to estimate construct reliability and the goodness of fit of the model. The average variance extracted (AVE) gives the percentage of manifest variables' variance retained by latent variables, where its value should exceed 0.5. In our model the AVE is somewhat lower than 0.5, but, according to Fornell and Larcker [43] (p. 46), "if the composite reliability (rho and alpha) of the three construct is well above the recommended level, the internal reliability of the measurement items is acceptable". AVE is 0.4564 for *Avoidance*, 0.4354 for *Stereotype*, and 0.3932 for *Positive emotional attitude*. In spite of the lower AVE, the measure of uni-dimensionality is acceptable, as construct reliability is strong. Figure 2 represents the final three-factor

model with indicator loadings and path coefficients. The χ2-test was not necessary, since the number of paths is equal to the number of variable pairs in the correlation matrix. Standardized root mean square residuals (SRMR) is an important model property that quantifies how strongly the empirical correlation matrix differs from the model-implied correlation matrix. The lower the SRMR, the better the theoretical model's fit. The Figure 2 model's SRMR is satisfactory 0.0627 ($d_{ULS}$ = 1.4874; $d_G$ = 0.3143). In ADANCO discriminant, validity is measured by HTMT, where the smaller the value of HTMT, the more likely the pairs of constructs are to be distinct. For the Figure 2 model, the HTMT values of each factor pair are under the proposed 0.85. Finally, construct reliability confirms the appropriateness of factor analysis. In ADANCO, construct reliability is measured by three indicators: Cronbach's alpha (*Avoidance* = 0.8549, *Stereotype* = 0.7800, *Positive emotional attitude* = 0.7379), Dijkstra–Henseler's rho (*Avoidance* = 0.9048, *Stereotype* = 0.7988, *Positive emotional attitude* = 0.7781), and Jöreskog's rho (*Avoidance* = 0.8935, *Stereotype* = 0.8412, *Positive emotional attitude* = 0.8127). All of the indicators of our model are highly satisfactory.

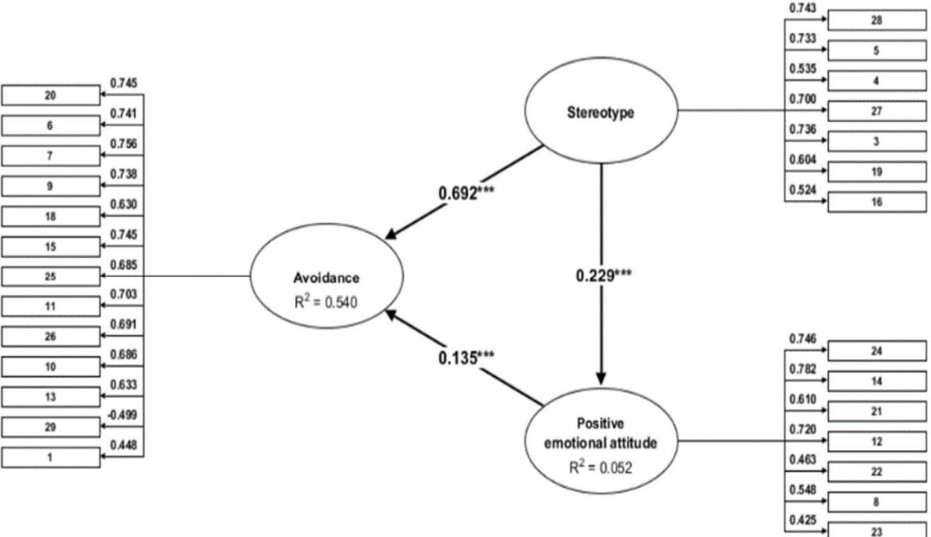

**Figure 2.** Structural model of FSA Note: *** $p < 0.001$.

Regarding overall model properties, discriminant validity, construct validity, and reliability, the Figure 2 three-factor model is an appropriate solution to examine inter-construct relationships. Our preliminary assumption was that *Stereotype* influences *Avoidance* and *Positive emotional attitude*; therefore, it became the independent construct. Since the *Avoidance* factor consists of statements reflecting expressive behavior, we decided to position it as a dependent variable. According to the meaning of *Positive emotional attitude* statements, this factor probably influences avoidance; thus, this became a dependent mediator construct. According to the path coefficients of Figure 2, every inter-construct relationship is significant. As Table 5 shows, the indirect effect of *Stereotype* (through *Positive emotional attitude*) on *Avoidance* is weak, but *Stereotype*'s direct effect is strong and substantial (Cohen's $f^2 > 0.8$). The direct effect of positive emotional attitude is significant but weak ($0.02 < f^2 < 0.15$).

The Figure 2 model was tested by gender, but the male sample's path coefficients were not significant. However, for females, every relationship was significant: *Stereotype* had a strong positive direct influence on *Avoidance* (path coefficient = 0.600 ***, R2 = 0.549), while the total effect with the *Positive Emotional attitude* was 0.7165 and strong (Cohen's $f^2$ = 0.5764). Regarding the respondents' age, an acceptable AVE (>0.4) was achieved for 30–40 year-old and 41–65 year-old adults. In the middle-aged adults' model, *Stereotype*'s direct effect on *Avoidance* was significant (path coefficient = 0.706 ***, $R^2$ = 0.590) and the total effect on *Positive Emotional attitude* was 0.7382 and very strong (Cohen's $f^2$ = 1.1911). The older adult's model showed *Stereotype*'s direct effect on *Avoid-*

*ance* as significant (path coefficient = 0.717 ***, $R^2$ = 0.552), and the total effect with the *Positive Emotional attitude* was 0.7330 and very strong (Cohen's $f^2$ = 1.129).

**Table 5.** Effect overview.

|  | β | Indirect Effect | Total Effect | Cohen's $f^2$ |
|---|---|---|---|---|
| Stereotype -> Avoidance | 0.6921 | 0.0308 | 0.7229 | 0.9863 |
| Stereotype -> Positive emotional attitude | 0.2285 |  | 0.2285 | 0.0551 |
| Positive emotional attitude -> Avoidance | 0.1348 |  | 0.1348 | 0.0374 |

## 5. Discussion and Conclusions

The aim of the current study was to investigate attitudes toward the older generation in a representative sample of the Hungarian population and to assess the reliability, validity, and psychometric properties of the Hungarian version of the Fraboni Scale of Ageism [29]. No research has yet used the full FSA scale to study ageism on a representative sample in an Eastern European country; therefore, our study has contributed to further testing the FSA scale in a different cultural setting. Since Hungary is an individualistic country [44], we hypothesized that the structure of the FSA ageism scale among the Hungarian population would resemble the three-factor structure found by Canadian, American, and other international studies. Our results provide support for the three-factor model of the FSA that has been found in previous studies.

The three factors identified by our research were named *avoidance, stereotypes, and positive emotional attitude*. The avoidance factor was similar to the avoidance factor of the original FSA [29], to the separation factor identified by Rupp et al. [34], and to the discrimination factor identified by Kutlu et al. [40]. This factor reflects respondents' inclination to avoid direct contact with old people. The stereotypes factor was similar to the antilocution factor of Fraboni et al. [33], and to the stereotypes factor of Rupp et al. [34], Kutlu et al. [40], and Fan et al. [39]. The stereotype factor describes respondents' stereotypical beliefs about old people. The third factor, *positive* emotional *attitude, was similar* to the *discrimination* factor identified by Fraboni et al. [33] and to *the affective attitude* factor found by Rupp et al. [34].

In the present study we also aimed to investigate age and gender differences that were found in previous international studies by using the FSA scale [33] to measure the degree of ageism. The results of the present study indicate that a significant negative relationship exists between respondents' chronological age and the scores of ageism. Men and younger respondents tend to be more ageist than older respondents. Regarding the type of settlement, the more positive image of older people among villagers is not justified by the age composition of the settlements, because, according to the data of the Central Statistical Office, [1] the proportion of people over 50 living in the capital and the villages is similar at around 38%. The explanation may be due to the more intense personal relationships among people in smaller settlements, allowing for a more personal experience instead of simply accepting stereotypes. The difference in attitudes between men and women is likely to be explained by the greater degree of empathy of women [45]. As the population of older adults continues to increase in Hungary, there exists a need for all members of society to possess an accurate knowledge of and positive attitudes toward this large group of adults that the young generation of today will also join some day in the future. In general, the results of our study provide additional support for the generalizability of the three-factor structure of the FSA scale found in previous studies. In conclusion, the present study proved that the FSA is a suitable instrument to measure discrimination against older people, and it offers additional evidence that the FSA is a reliable, valid, and multidimensional measure of ageism.

## 6. Limitation and Implication for Further Studies

The limitation of our research is the online data collection, which excluded those people who are not active online. In the future, it would be worthwhile to ensure the wider presence of the Hungarian population in the survey by means of hybrid data collection.

Further testing of the scale would require a repetition of the data collection, ideally in a longitudinal form.

Rupp et al. [34] and other scholars, e.g., Ayalon et al. [25] and Klusmann et al. [40], suggested that it would be fruitful for future research to study the construct of ageism in a more diverse group of countries. The present study represents one step in this direction. In addition, future cross-cultural studies are recommended, i.e., in other post-socialist countries of Eastern Europe, where over 60% of the population, e.g., Romania and Bulgaria, [3] admitted that ageism exists in their countries. Lastly, more research would be needed to explore which additional variables, such as cultural values, knowledge, anxiety of aging, or health, might mediate or moderate the effects of ageism.

**Author Contributions:** Conceptualization, Á.H.-T. and Á.N.; methodology, Á.N. and J.D.; data analysis, Á.H.-T. and J.D.; writing—original draft, Á.H.-T., writing—review and editing, Á.H.-T., Á.N. and J.D. All authors have read and agreed to the published version of the manuscript.

**Funding:** This research received no external funding.

**Institutional Review Board Statement:** Not applicable.

**Informed Consent Statement:** Not applicable.

**Data Availability Statement:** Data sharing is not applicable. The data are not publicly available due to participants' privacy.

**Conflicts of Interest:** The authors declare no conflict of interest.

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
