# Peer review of "Measuring Discrimination against Older People Applying the Fraboni Scale of Ageism"

_information, doi:10.3390/info12110458_

Round 1
Reviewer 1 Report
The manuscript describes the adaptation of the full Fraboni Scale of Ageism (FSA) to Hungarian and assessment of its reliability, validity, and psychometric properties. The rationale and importance of the study are well described. In addition, the factor structure and construct validity of the FSA are well structured and presented in systematically and clearly. However, the theoretical background of the study, especially the definition of ageism, is often vague and insufficiently coherent and clear, as detailed below. 1. In the theoretical background, the distinction between prejudice and stereotypes is not clear enough. On page 2, line 57, stereotypes are represented as having a similar meaning ("There are prejudices (stereotypes) about old age in some societies that old age is a single stage at the end of life when people are mostly sick, weak, poor, and lonely"), yet, in the following sentence (Page 2, line 59) they appear separately, which implies different meanings for each term ("The consequence of such similar prejudices and stereotypes is age discrimination, called ageism "). The research literature tends to distinguish between prejudice, stereotypes and discrimination, where prejudice is related to the affective component, stereotype to the cognitive component, and discrimination to the behavioral component (see for example- Iversen, T. N., Larsen, L., & Solem, P. E. (2009). A conceptual analysis of ageism. Nordic Psychology, 61(3), 4-22.). 2. In addition, the definition of ageism that appears in the first paragraph of the subchapter "The concept of Ageism"(page 2, lines 58-62) relates mostly the discrimination component, while it is not clear enough that the two other components, stereotypes and prejudices, are also included in the definition of ageism (as is detailed in the next paragraph). The fuzziness between ageism and age discrimination is also reflected in the third paragraph of this subchapter (page 2, lines 74-82). 3. In the second paragraph of the subchapter " 2.2. Measurement of Ageism" (page 3 lines 101-105), the definition of implicit ageism is not clear enough. It is also recommended to give examples of scales that assess implicit ageism. 4. In this subchapter, it is recommended to detail what is the only scale in the study of Ayalon et al. to meet minimum requirements for psychometric validation (page 3 lines 105-107). 5. In the subchapter "3.1. Data Collection and Sample", it is recommended to expand and explain more about the pilot study (page 4 lines 173-178): Who were the respondents? What validation method was used?Author Response
Dear Reviewer,
Please see the attachment!
Best regards
Agne Hofmeister-Tóth

Reviewer 2 Report
When addressing ageism and negative attitudes about aging it is imperative that we, as researchers and authors, not inadvertently contribute to ageist discourse. To that end the opening sentence (The aging of the world’s population today is one of the greatest challenges of the 21st 26
century) positions aging as a problem and a challenge. This is problematic for several reasons. First - aging is not about older people we are all aging and classifying it as an older person issue creates an us vs them mentality which promotes ageism. Second, aging is not a problem or a challenge, it is an opportunity. Ageism is the greatest challenge of the 21st century. I suggest that the authors re-position this argument in the opening paragraph.
2. Page 2, line 63 - ageism is misspelled
Author Response
Dear Reviewer,
Thank you very much for your review!
We fully agree with your comment and revised the sentence:
"The current aging of the world’s population is the 21st century's dominant demographic phenomenon, which is unprecedented in human history."
We hope you will accept this version of the sentence.
Best regards
Agnes Hofmeister-Tóth
Round 2
Reviewer 1 Report
I confirm all the corrections, I have no additional comments.